

# Response surface methodology for the mixed fungal fermentation of *Codonopsis pilosula* straw using *Trichoderma reesei* and *Coprinus comatus*

Ti Wei[1,2], Hongfu Chen[1,2], Dengyu Wu[2,3], Dandan Gao[1,3,4], Yong Cai[1,2,3], Xin Cao[2,4], Hongwei Xu[1,4], Jutian Yang[1,2,4] and Penghui Guo[1,2,3]

[1] College of Life Sciences and Engineering, Northwest Minzu University, Lanzhou, Gansu, China
[2] Ecological Industry Development Research Institute of the Upper Yellow River, Northwest Minzu University, Lanzhou, Gansu, China
[3] Taizishan Ecosystem Observatory of Carbon Neutralization, Northwest Minzu University, Lanzhou, Gansu, China
[4] Institute of Livestock and Poultry Genetic Resources Conservation and Utilization of the Upper Yellow River, Northwest Minzu University, Lanzhou, Gansu, China

Corresponding author
Penghui Guo,
skyguoph@xbmu.edu.cn

## ABSTRACT

The objective of this study was to investigate the cellulose degradation rate (CDR) and lignin degradation rate (LDR) of *Codonopsis pilosula* straw (CPS) and the optimal fermentation parameters for mixed fungal fermentation. Single-factor tests were used to study the effects of the fungal ratio (*Trichoderma reesei*: *Coprinus comatus*), fungal inoculum, corn flour content, and fermentation time on the degradation rate of cellulose and lignin. Based on the results of this experiment, the optimal fermentation factors were identified, and the effects of various factors and their interactions on the degradation rates of cellulose and lignin were further evaluated using the response surface method. The quadratic polynomial mathematical model of degradation rates of the cellulose and lignin in CPS by mixed fungus fermentation was established using Design Expert software v8.0.6. Under the optimal parameters for fungal fermentation of CPS straw (fungal ratio 4:6, fungal inoculum 8%, corn flour content 10%, fermentation time of 15 d), the CDR and LDR reached 13.65% and 10.73%, respectively. Collectively, the mixed fungal fermentation of CPS resulted in decreased lignin and cellulose content, better retention of nutrients, and enhanced fermentation quality. The results of this study indicate that fermentation using *Trichoderma reesei* and *Coprinus comatus* is a productive method for straw degradation, providing a theoretical basis for the development of CPS as feed.

## INTRODUCTION

Lignocellulose is the richest renewable and low-cost biomass energy in the world. Around 10 billion tons of lignocellulose is produced annually, about 70% of which is crop straw because of its high yield and considerable resource potential (*Sánchez Óscar & Cardona,*

*2008*). Despite such an enormous number of straw resources, many developing countries do not maximize the potential of crop straw (*Wang et al., 2018*). Straw is a very difficult material to break down because of its crystal structure, composed of hemicellulose, lignin, and cellulose. Across the world, large amounts of straw are directly burned, stacked, discarded, or buried, which is a waste of resources and a significant cause of environmental pollution. It is common to use the microbial degradation method on wheat, corn, and other crop straws, but there is little is research on CPS resources (*Zhao et al., 2020*). With the roots of *Codonopsis pilosula* having a large pharmacological value, the CPS industry is focused on planting, large-scale cultivation, and product development, and does not divert resources to research or the utilization of the large amount of straw resources generated during the production process. Many straw resources are discarded as waste or burned on site, which is both harmful and wasteful, so effective uses for straw resources are needed.

*Codonopsis pilosula* is a Chinese herbal medicine that is cultivated worldwide, particularly in China, Japan, and Korea (*Chen et al., 2018a*; *Liu et al., 2018*). The cultivation area of *Codonopsis pilosula* has significantly increased to match an increase in market demand. Tangshenosid, codonopsine, neochorogenic acid, and lobetyolin are universal active substances in *Codonopsis pilosula* (*Gao et al., 2018*). These substances have characteristics that have been shown to strengthen the spleen, moisten the lungs, nourish blood, engender liquid, regulate and enhance immune functions, and promote anti-tumor activity (*Bai et al., 2018*; *Zou et al., 2014*). Generally, the roots of *Codonopsis pilosula* are used as medicine, but the above-ground portions, including the leaves, stems, and flowers, are discarded (*Fu et al., 2018*).

A previous study reported that the dry matter content of *Codonopsis pilosula* straw (CPS) reached 91.74%, including 17 different amino acids and nitrogen-free extracts (31.63%), which made it a valuable and feasible ruminant feed resource (*Ren et al., 2015*). It has also been shown that using CPS as a dietary supplement can increase feed palatability, body resistance, and feed conversion rate, promoting pig growth and fattening, and improving the laying rate of layers (*Hong et al., 2019*). Lignin is a refractory polymer encasing cellulose and hemicelluloses and impeding their utilization by rumen microorganisms, so CPS has low digestibility and feed values for ruminants (*Van Kuijk et al., 2016*). Numerous experiments have been conducted to increase the digestibility of crop straw by degrading lignocellulose structure (*Zhang et al., 2018*), and the pretreatment of high lignocellulose forage using biotechnology has received considerable attention in recent years. Under specific temperature and humidity conditions, cellulose-degrading lactic acid bacteria and other anaerobic microorganisms degrade the cellulose and hemicellulose in crop straw during anaerobic fermentation (*Li et al., 2019*; *Li et al., 2020*). This produces an abundance of acid, requiring the pH value to be further reduced to inhibit the growth of spoilage bacteria (*Nayan et al., 2018*; *Dagar et al., 2015*). Applying anaerobic fermentation technology to straw reduces the hemicellulose and cellulose content, improves the digestibility of the straw, prevents a loss of nutrients, improves the fermentation quality and *in vitro* digestibility, enhances the palatability of straw, and increases its nutritional value (*Sun et al., 2019*). Diverse biological approaches have been used to study wheat, corn, and other crops, but little research has been conducted on their

use in Chinese medicine (*Brodeur et al., 2011*; *Raghuwanshi, Misra & Saxena, 2014*). The total ban on the use of antibiotics as growth promoters in animals was introduced in the European Union on 1 January 2006 (Regulation (EC) No, 1831/2003; *Szuba-Trznadel et al., 2021*). This led to a gradual, worldwide ban on antibiotic feed additives to ensure the healthy and sustainable development of animal husbandry. China banned antibiotics as a growth promoter and prohibited the prophylactic use of antibiotics in feed in 2020 (except in traditional Chinese medicine). Therefore, developing a green, safe, and healthy antibiotic alternative with no side effects has become an important issue in feed production. Chinese medicine is considered one of the potential alternatives to antibiotics. Using the non-medicinal, unused plant parts from the production process of traditional Chinese medicine resources as raw feed materials to develop new feeds would save straw resources from being wasted and help meet the increasing demand for animal feed products. This study used lignocellulose-degrading bacteria, such as mold and white rot fungi, to ferment and prepare high-quality fermented forage (*Zhao et al., 2020*; *Sánchez, 2009*; *Kogo et al., 2017*). Cellulase and laccase were produced by *Trichoderma reesei* and *Codonopsis pilosula*, which can break down lignocellulose in CPS. *Trichoderma reesei* and *Codonopsis pilosula* are reported to grow synergistically, increasing enzyme production efficiency (*Ge et al., 2009*).

The objective of the present study was to determine if CPS was able to be developed as feed. This study tested the following hypotheses: (i) the lignocellulose in CPS could be degraded by *Trichoderma reesei* and *Coprinus comatus* and (ii) the optimal CPS fermentation conditions could be identified using response surface methodology (RSM), providing guidance for preparing high-quality CPS-fermented feed.

## MATERIALS AND METHODS

### Test materials

#### Fungal strains and spawn preparation

The *Trichoderma reesei* (strain CGMCC 3.5218) and *Coprinus comatus* (strain CC900) used in this study were purchased from the China General Microbiological Culture Collection Center (Beijing, China) and the Tianda Institute of Edible Fungi (Jiangsu, China), respectively. Before beginning the experiment, two strains of fungi were resuscitated according to the protocols of our research team. Two fungi were maintained on potato dextrose agar plates containing (g/L): potato 200, dextrose 20, and agar 15, and the pH value was not adjusted. The fungi were grown in a 28 °C constant temperature incubator for roughly a week until mycelia colonized and completely covered the agar plates. Finally, the fungi were stored at 4 °C and subcultured every 3 months.

#### Codonopsis pilosula straw

CPS was collected at the ripening stage (dry matter, 90%) from Dingxi City, Gansu Province (34.26–35.35 N, 103.52–105.13 E). The collected CPS was then chopped into approximate lengths of 2–3 cm and then crushed into granules with an average particle size of 0.2 cm using a pulverizer (93ZR-8.0C, Shengtailong Mechanical Equipment Company, Feicheng, China) to facilitate subsequent fermentation. These granules were

**Table 1 Chemical composition (g/kg DM) of *Codonopsis pilosula* straw.**

| Items | Content (g/kg) |
|---|---|
| Dry matter | 905.7 ± 16.32 |
| Crude ash | 70.8 ± 2.21 |
| Ether extract | 48.4 ± 0.92 |
| Crude protein | 154.8 ± 3.55 |
| Acid detergent fiber | 541.8 ± 8.24 |
| Neutral detergent fiber | 738.7 ± 9.20 |
| Cellulose | 417.6 ± 11.36 |
| Lignin | 114.6 ± 7.25 |

stored in sealed plastic bags in the laboratory and kept in a cool, dry room to avoid deterioration and rotting. The chemical compositions of CPS are presented in Table 1.

### Preparation of liquid fermentation broth

The liquid fermentation medium was improved and prepared based on the methods described by *Kogo et al. (2017)* using S. *pastorianus*. The medium was composed of 200 g/L potatoes, 20 g/L glucose, 2 g/L anhydrous ammonium sulfate, 1 g/L peptone, 1 g/L anhydrous potassium dihydrogen phosphate, and 1 g/L anhydrous magnesium sulfate, which was cultured for 2 d to *Trichoderma reesei* and 5 d to *Coprinus comatus* in a rotary shaker (150 rpm) at 28 °C. To guarantee that the spore concentration was constant in this investigation ($1 \times 10^7$ cells/mL), the hemocytometer plate was used to count the spore concentrations of the two strains after culture (*Chen et al., 2018b*; *Chen, Yan & Xu, 2011*; *Xu et al., 2017a*).

## Experimental design for *Codonopsis pilosula* straw fermentation

### Substrate preparation

The substrate included 90% CPS and 10% corn flour. The substrate and water were well mixed in a 2:3 weight basis ratio and compacted into self-sealing bags. (17 cm × 25 cm).

### Single-factor inoculation of Trichoderma reesei and Coprinus comatus

Using fungus ratio (*Trichodemma reesei*: *Coprinus comatus*), mixed fungal inoculum, corn flour content, and fermentation time as single factors, the effects on the degradation rate of lignin and cellulose in the CPS were investigated, with three replicates in each group.

Group 1: Testing for best fungal ratio: the fungal liquid was inoculated with *Trichodemma rcesei* and *Coprinws comatus* with ratios of 10:0, 7:3, 5:5, 3:7, and 0:10 in the substrate. The mixed fungal inoculum was 10% (water content basis), corn flour content was 10% (weight basis), and fermentation time was 10 d.

Group 2: Testing for best fungal inoculum: mixed fungal inoculums of 5%, 10%, 15%, and 20% (water content basis) were added to the substrate, using the optimal fungal ratio determined from Group 1 and same proportions and fermentation time as Group 1.

Group 3: Testing for optimal nutrient additive amount: 5%, 10%, 15%, and 20% (weight basis) of corn flour was added into the substrate, using the optimal fungal ratio determined

in Group 1, the optimal fungal inoculum determined from Group 2, and a 10-day fermentation time.

Group 4: Testing for optimal fermentation time: Using the optimal fungal ratio, fungal inoculum, and nutrient additive amounts determined in Groups 1-3, the fermentation was carried out on the substrate for 5 d, 10 d, 15 d, and 20 d.

### Conditional optimization of fermentation

The effects of numerous parameters and their interactions on the breakdown of cellulose and lignin in corn straw were evaluated using the response surface approach. According to the design principle of the Box-Behnken experiment, four factors were selected, including fungal ratio, mixed fungal inoculum, corn flour content, and fermentation time, while CDR and LDR were taken as response values (Design Expert 8.0.6). An experimental response surface analysis was conducted with three levels of each factor. A mathematical model was established between the degradation rate of lignin and cellulose and the various factors tested and these calculations were used to determine the optimal conditions for using *Trichoderma reesei* and *Coprinus comatus* mixed fermentation to break down lignocelluloses (Table S1).

### Characterization of Codonopsis pilosula straw

Characterization scanning electron microscope (SEM) imaging was carried out using a Gemini 500 SEM. The chemical phase analysis was carried out using a PANalytical X-ray diffractometer (XRD), which was equipped with CuKα radiation (λ = 0.15406 nm). Fourier-transformed infrared (FTIR) spectra were obtained from an FTIR spectrometer (FTIR-7600, Lambda Scientific, Edwardstown, SA, Australia).

### Chemical analyses

Each sample's neutral detergent fiber (NDF), acid detergent fiber (ADF), and lignin were measured after the single-factor test. Dry matter (DM), crude protein (CP), ether extract (EE), ash, NDF, ADF, lignin, and volatile fatty acids were evaluated in the samples before and after response surface methodology optimization. After drying the samples to a constant weight at 65 °C, the DM content of the samples was calculated. CP content was calculated by multiplying nitrogen content by 6.25. The diethyl ether extract was determined using the Soxhlet method (*Wang et al., 2021*). After the samples were heated to 600 °C for 6 h in a muffle furnace, the amount of ash was measured. The ash-free NDF (*VanSoest, Robertson & Lewis, 1991*), ash-free ADF, and lignin were determined as previously described. We then calculated the difference between NDF and ADF to determine the hemicellulose content, and the difference between ADF and acid detergent lignin (ADL) to determine the cellulose content (*VanSoest, 1973*). A total of 20 g of uniformly mixed fermented feed was placed in a juicer, and 180 mL of distilled water was added, homogenized for 2 min, and filtered with four layers of gauze to prepare a crude extract. This was then filtered with double-layer filter paper and left standing for 0.5 h. Analysis of organic acids was carried out by high-performance liquid chromatography (HPLC). After adequate dilution, the samples were filtered through a 0.2 μm polypropylene filter before analysis in a Dionex UltiMate 3000 RSLC HPLC system

(Thermo Fisher Scientific, Waltham, MA, USA) connected to a Shodex RI-101 differential refractive index detector (Showa Denko, Tokyo, Japan). The organic acids were separated using the Aminex HPX-87H analytical column coupled to a guard column (Bio-Rad, Hercules, CA, USA). A series of standard solutions, including acetic acid, lactic acid, butyric acid, and propionic acid, were prepared and analyzed together with the samples. The temperature was set to 30 °C and the mobile phase consisted of a methanol solution of 10% and 0.1% v/v phosphoric acid solution of 90% with a flow rate of 0.6 ml/min (*Huang et al., 2021*).

## Statistical analyses

Experimental data, including the chemical compositions of CPS after different treatments, were expressed as means and analyzed by IBM's Statistical Product and Service Solutions software (SPSS 21.0). ANOVA was conducted to evaluate differences, followed by a Duncan's multiple range test. Differences were considered statistically significant at $P < 0.05$.

## RESULTS

### Effect of single-factor inoculation of *Trichoderma reesei* and *Coprinus comatus* on the degradation rate of cellulose and lignin in *codonopsis pilosula* straw

#### Influence of the fungal combination ratio

As the proportion of *Trichoderma reesei* gradually decreased and *Coprinus comatus* gradually increased (Fig. 1A), cellulose and lignin also decreased gradually. Using a 3:7 ratio (*Trichoderma reesei*: *Coprinus comatus*), the CDR and LDR reached 12.3% and 5.67%, respectively, with an initial transient increase followed by a slight decrease in degradation rate, verifying that *Coprinus comatus* is important to lignin decomposition (Fig. 1B). When *Trichoderma reesei* was not included in the substrate (0:10 ratio), the CDR decreased.

#### Influence of fungal inoculum

The degradation rates of cellulose and lignin increased as the inoculum size increased (Fig. 2A). The peak CDR (13.22%) and LDR (8.44%) were observed using an inoculum size of 10% (Fig. 2B).

#### Influence of cornmeal amounts

At a corn flour concentration of 10%, the cellulose and lignin content was the lowest, and peak CDR (13.30%) and LDR (10.47%) were observed (Figs. 3A and 3B). The degradation then fell, marginally, as the corn flour concentration increased ($P < 0.05$). Batch fermentation was conducted to save costs, and the optimal addition of corn flour was set to 10%.

#### Influence of fermentation duration

*Trichoderma reesei* and *Coprinus comatus* primarily grew hyphae in the first 5 d, but the CDR and LDR increased significantly over time. The degradation rates of cellulose and

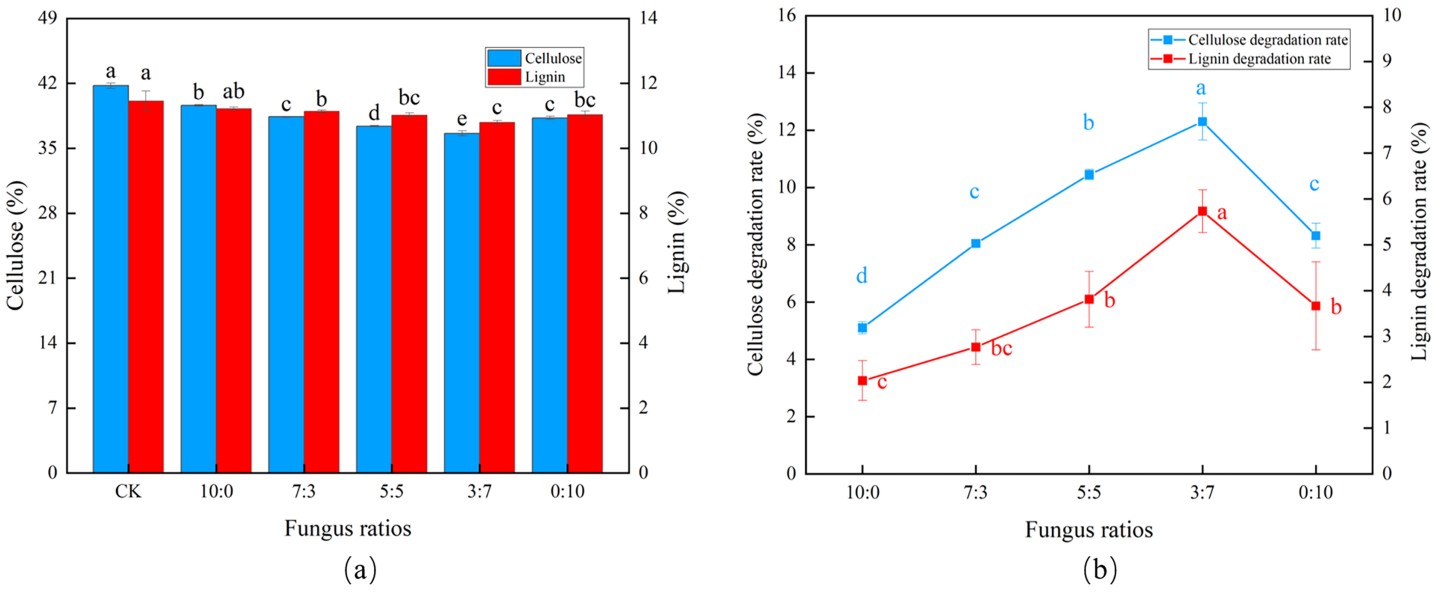

**Figure 1 Effects of different fungal ratios on the degradation of cellulose and lignin in *Codonopsis pilosula* straw.** (A) Cellulose and lignin content, (B) cellulose and lignin degradation rate.           

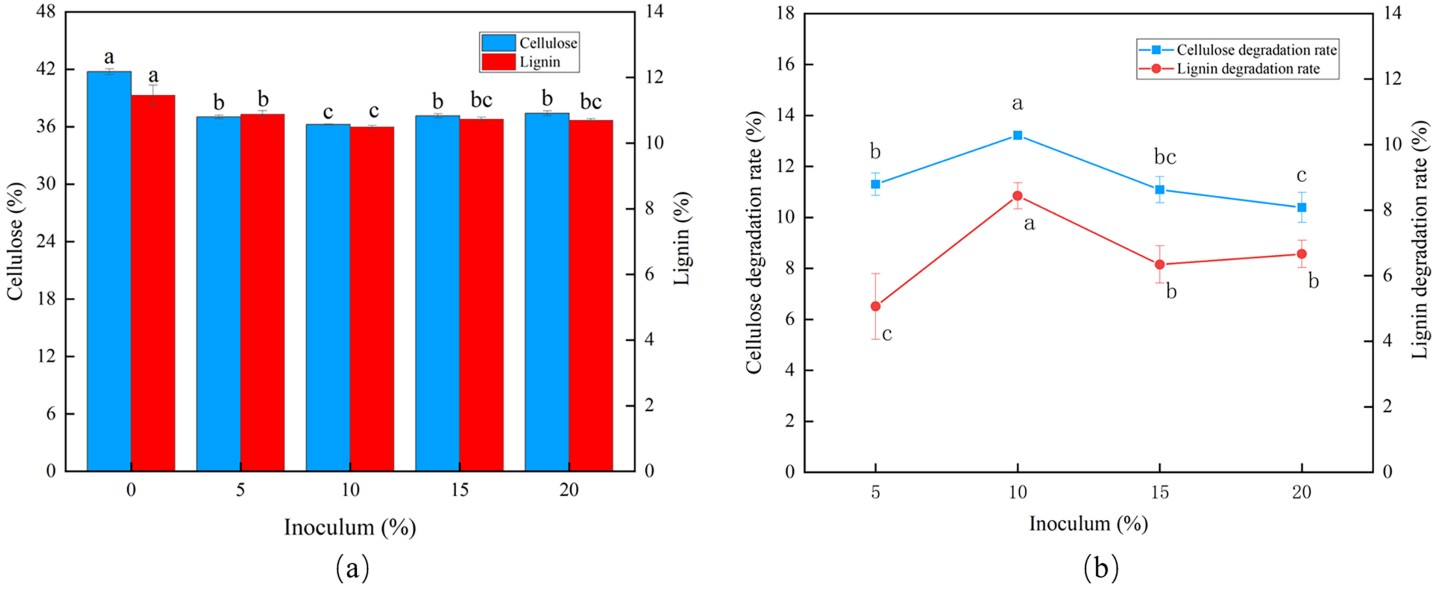

**Figure 2 Effects of different inoculum on the degradation of cellulose and lignin in *Codonopsis pilosula* straw.** (A) Cellulose and lignin content, (B) cellulose and lignin degradation rate.           

lignin decreased after 10 d of fermentation, and after 15 d, the CDR and LDR were 13.61% and 9.33%, respectively (Figs. 4A and 4B), indicating the optimal fermentation duration was 15 d.

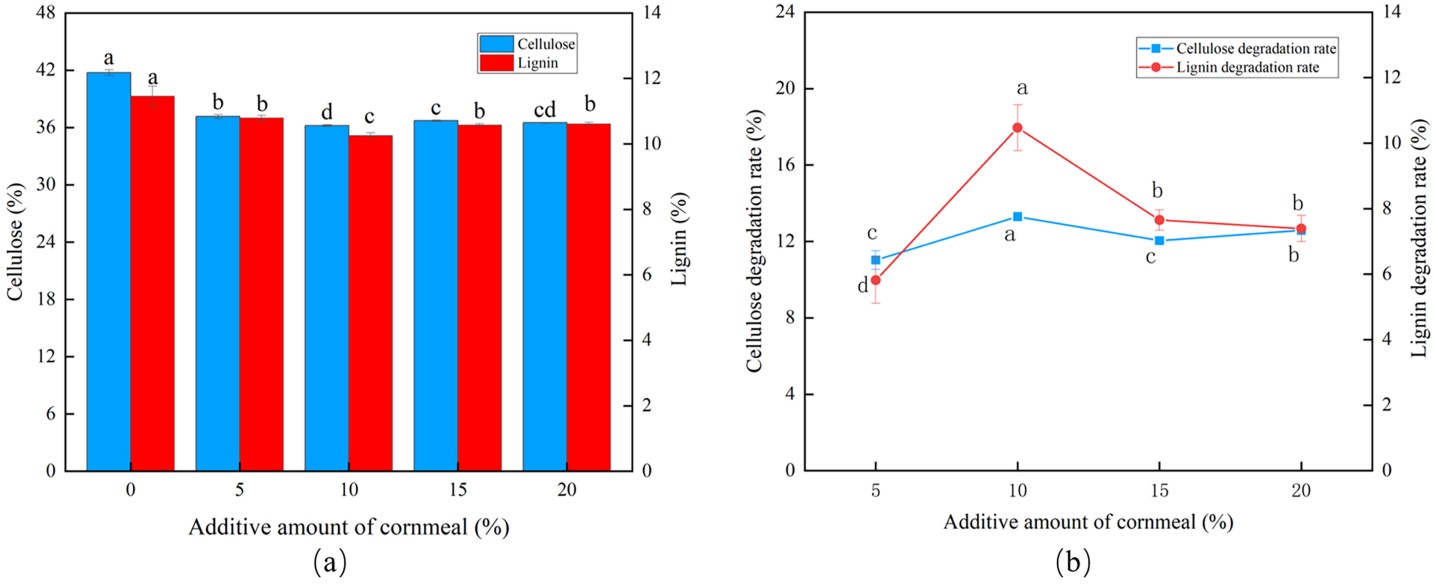

**Figure 3 Effects of different corn flour amounts on the degradation of cellulose and lignin in *Codonopsis pilosula* straw.** (A) Cellulose and lignin content, (B) cellulose and lignin degradation rate.

## Optimization of fermentation conditions of *Trichoderma reesei* and *Coprinus comatus* using response surface methodology
### Response surface experimental design and result analysis

Design Expert software was used to further optimize the parameters of each factor. A, C, D, AB, AC, CD, A², B², C², and D² had highly significant effects on the CDR ($P < 0.01$), and C and AD had significant effects on the CDR ($P < 0.05$), indicating that these were solid (Table S2). No significance was found for the missing items, indicating that there were no abnormal points in the test data, and that the model was appropriate (Table S3). Moreover, A, B, C, D, AB, BC, B², C², and D² had highly significant effects on LDR ($P < 0.01$), which indicates that these were important factors in the solid-state fermentation process. The missing items were not significant, indicating that there were no abnormal points in the test data and that the model was appropriate (Table S4).

According to the regression analysis of the response surface coefficient of CDR, the fitting equation of the model was: $Y = 9.32A + 3.66B + 5.45C + 11.91D - 0.95AB + 0.56AC + 0.47AD + 0.07BC - 0.04BD + 0.39CD - 1.08A2 - 0.32B2 - 0.37$. The regression model was highly significant ($P < 0.01$), while the lack-of-fit test was not significant ($P = 0.8196$), indicating that the regression equation had a good fitting degree. The regression equation's coefficient of determination ($R^2$) was 0.93, indicating that this model can explain 93% of the variation in CDR, suggesting that it is a good fit for actual data (Table S3).

According to the regression analysis of the response surface coefficient of LDR, the fitting equation of the model was: $Y = 4.25A + 8.59B - 0.46C + 5.54D - 0.95AB - 0.17AC + 0.11AD + 0.30BC + 0.04BD + 0.15CD - 0.08A2 - 0.24B2$. Consistently, the regression model was highly significant ($P < 0.01$), and the lack-of-fit test was not significant ($P = 0.2428$), which indicates that the regression equation had a good fitting degree. The $R^2$
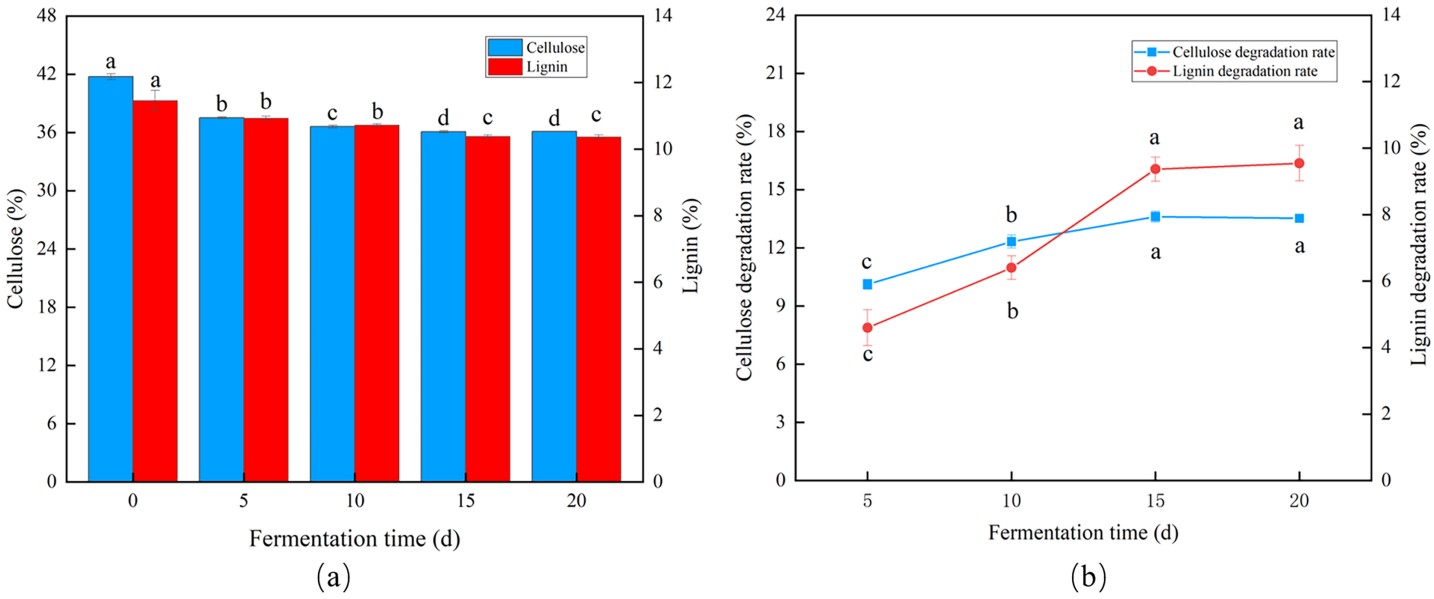

**Figure 4 Effects of different fermentation times on the degradation of cellulose and lignin in *Codonopsis pilosula* straw.** (A) Cellulose and lignin content, (B) cellulose and lignin degradation rate.

of the regression equation was 0.93, indicating that 93% of the variation in CDR could be explained by this model and that the model is a good fit for actual data (Table S4).

### Response surface analysis of cellulose degradation rate

Based on the regression equation, the response surface plot was generated to directly reflect the influence of each element and the interaction of two factors on the response value.

The steeper the surface slope in the interactive graph, the closer the contour line was to the ellipse, suggesting greater influence of this factor on the response value. Two factors were fixed, and the interaction between the other two factors was analyzed. When the fungal ratio increased, and the fungal inoculum decreased, the CDR increased, and the trend slope on the cross plot was high, indicating that these two factors significantly influenced the CDR (Fig. S1B). The cellulose degradation rate exhibited an upward trend with an increase in fungal ratio and corn flour content (Fig. S2B). With an increased fungal ratio and fermentation time, the CDR also increased, and the slope of the interaction graph between them was high, indicating a significant influence on the CDR (Fig. S3B). The contour lines of the above interaction diagrams were all elliptical, indicating significant interactions, consistent with the results of the variance analysis (Fig. S4B).

### Response surface analysis of lignin degradation rate

With increased fungal ratio and inoculum size, the LDR exhibited a downward trend, indicating that both significantly influenced the LDR (Fig. S5B). The LDR decreased with increased inoculum and exhibited a transient increase followed by a decrease as the amount of corn flour added increased (Fig. S6B). The contour lines of the above interaction diagrams were all elliptical, indicating significant interactions, consistent with the variance analysis results.

**Table 2 The chemical composition of fermented *Codonopsis pilosula* straw under the optimum conditions.**

| Items | Content (g/kg) |
|---|---|
| | Fermented |
| Dry matter | 334.7 ± 11.76 |
| Crude ash | 72.4 ± 2.21 |
| Ether extract | 63.7 ± 1.91 |
| Crude protein | 153.6 ± 1.61 |
| Acid detergent fiber | 473.3 ± 4.77 |
| Neutral detergent fiber | 683.8 ± 5.93 |
| Cellulose | 360.6 ± 8.10 |
| Lignin | 102.3 ± 3.62 |

### Response surface optimization results and model verification

The optimal conditions for the simultaneous degradation of cellulose and lignin, obtained using Design Expert software, were as follows: a fungal ratio of 4:6, inoculum size of 8%, a corn flour content amount of 10%, and a fermentation duration of 15 d. Under these conditions, the predicted CDR and LDR were 14.37% and 11.65%, respectively.

The optimal conditions of model optimization were tested to verify the model's effectiveness. The solid-state fermentation test was carried out, and the verification test was set up with three replicates, with the average degradation rate of cellulose and lignin being 13.65% and 10.73%, respectively. These results indicate that this model can predict the degradation of cellulose and lignin. Other chemical components of fermented CPS were then determined using this model. The CP increased significantly, ADF, NDF, cellulose, and lignin decreased significantly, while crude ash and CP did not change significantly (Table 2).

### Fermented Codonopsis pilosula straw was characterized through XRD, FTIR, SEM, and XRD analyses of Codonopsis pilosula straw

The crystallinity of cellulose in *Codonopsis pilosula* straw stem before and after fermentation was analyzed in an XRD experiment, as shown in Fig 5. The two well-defined crystalline peaks represent amorphous at around $2\theta = 22°$ and crystalline peaks $2\theta = 27°$. The peak values of crystal planes $2\theta = 22°$ and $2\theta = 27°$ in the fermentation treatment group were significantly lower than those in the control group, indicating that the crystallinity of the straw after treatment with *Trichoderma reesei* and *Coprinus comatus* was lower than that in the control group. As expected, the magnitude of these crystalline peaks reduced upon hydrolysis, which successfully eliminated the non-cellulosic materials in the amorphous regions of the *Codonopsis pilosula* straw stem. The fermentation treatment increased the amount of cellulose exposed on the fiber surface, thereby increasing surface roughness.

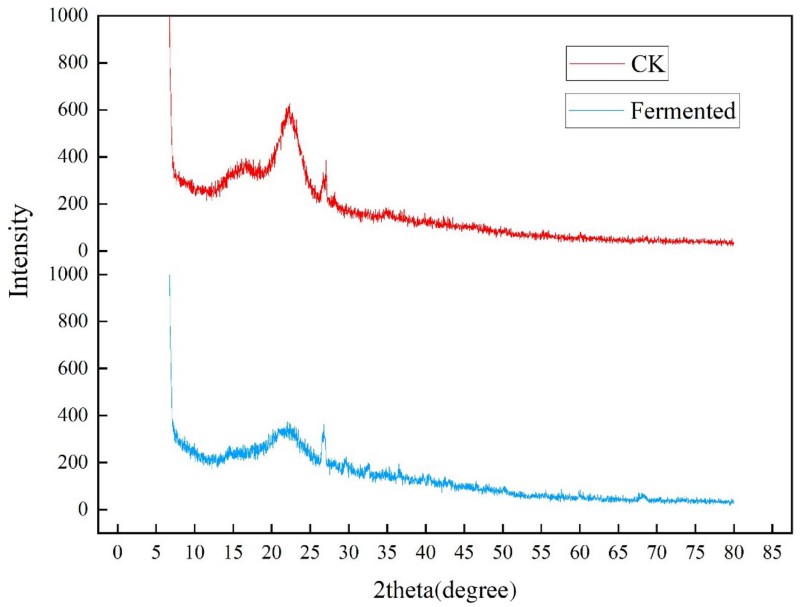

**Figure 5  XRD diffraction results of *Codonopsis pilosula* straw.**

## FTIR analysis of Codonopsis pilosula straw

To further investigate the influence of *Trichoderma reesei* and *Coprinus comatus* on the structure of *Codonopsis pilosula* straw, their chemical structures were characterized using a Fourier-transformed infrared (FTIR) spectrometer (Fig. 6). Figure 6 shows that, compared with the untreated *Codonopsis pilosula* straw, the absorption of the fermented *Codonopsis pilosula* straw by *Trichoderma reesei* and *Coprinus comatus* decreased at 1,650 and 1,429 cm$^{-1}$ (the characteristic absorption peak of benzene ring), indicating that lignin was degraded to a certain extent. The peak of 1,201–1,357 cm$^{-1}$ (C-O stretching vibration in lignin) represents the byproducts of lignin decomposition, such as phenol, ether, alcohol, and ester. The increase in the intensity of these peaks indicates an increase in these byproducts. After pretreatment, the peak at 1,022 cm$^{-1}$ (C-O stretching vibration, carbohydrate (cellulose, hemicellulose) or polysaccharide) was enhanced, showing that cellulose is degraded into sugars. The 3,300–35,00 cm$^{-1}$ wide absorption band corresponds to carbohydrates (cellulose, hemicellulose, starch, monosaccharide, *etc.*). The decrease of the peak here indicates that the stretching vibration of the hydroxyl group was strengthened. This may be because the ether bond broke, making the molecular structure looser, exposing the internal hydroxyl group, and increasing the hydroxyl group of lignin. This shows that the lignin undergoes a demethylation reaction, which can reduce the steric hindrance inside the lignin and improve its reaction activity. These results show that the chemical structure of the cellulose and lignin of the *Codonopsis pilosula* straw were destroyed by the introduced strains, effectively degrading the cellulose and lignin.

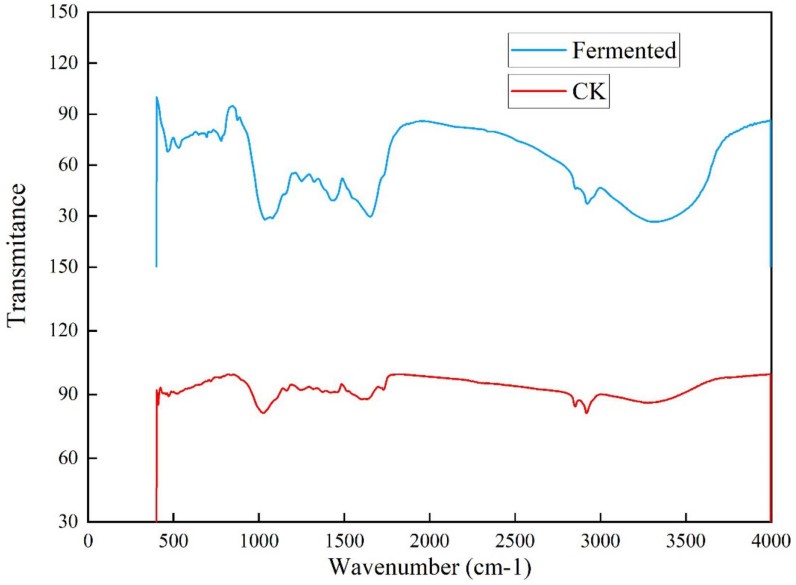

**Figure 6 FT-IR results of *Codonopsis pilosula* straw.**

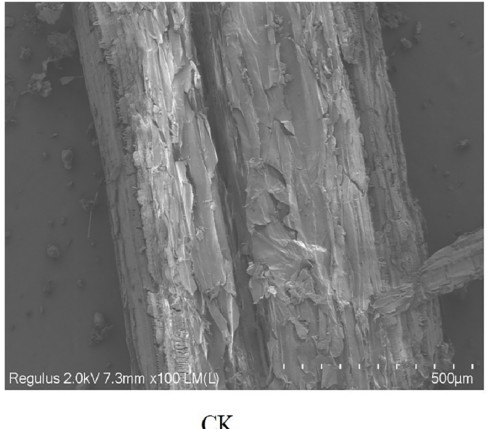

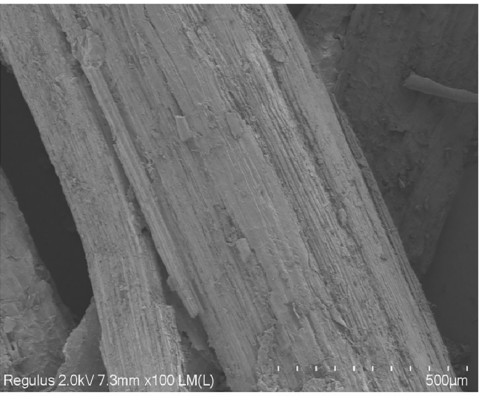

CK                    Fermented

**Figure 7 SEM images of *Codonopsis pilosula* straw.**

### SEM analysis of Codonopsis pilosula straw

The morphological structures of both raw and fermented *Codonopsis pilosula* straw samples were examined through a scanning electron microscope (SEM) and are presented in Fig. 7. The surface of the straw in the control group, as shown in Fig. 7, is relatively smooth, covered with amorphous components, and has a relatively complete structure. After adding two kinds of *Trichoderma reesei* and drumstick fungi for fermentation, the straw surface showed corrosion and tearing, with the straw surface becoming loose, with exposed cellulose and lignin, and broken and rough fiber bundles. Lignocellulose-degrading enzymes secreted by *Trichoderma reesei* and *Coprinus comatus* during the growth process led to some degradation in *Codonopsis pilosula* straw. With the shedding of

the primary cell wall, more inferior walls are exposed, which is conducive to the in-depth role of enzyme molecules.

## DISCUSSION

### Effect of *Trichoderma reesei* and *Coprinus comatus* on the degradation rate of cellulose and lignin in *Codonopsis pilosula* straw

*Codonopsis pilosula* is a traditional medicine that has been used worldwide for many decades (*Gao et al., 2018*). Nevertheless, *Codonopsis pilosula* straw (CPS) has been largely understudied and is often burnt as waste. Few studies have examined the use of CPS as crop straw feed. It has been established that straw is primarily made up of cellulose, hemicellulose, and lignin. Biodelignification can be done using microbes, as these microorganisms produce enzymes to attack, and degrade polymers in lignocellulosic substrates. Many microorganisms in nature can degrade lignin, including fungi and bacteria. This study used the microbial fermentation method to ferment and degrade the lignocellulose of CPS, specifically using the fungal microorganisms *Trichoderma reesei* and *Coprinus comatus*. Genetically-engineered *Trichoderma reesei* is one of the most promising fungi for cellulase production, with an estimated 60 to 100 g of extracellular protein per liter of cultured *Trichoderma reesei* (*Ike et al., 2010*). *Coprinus comatus* reportedly produces ligninolytic enzymes, which delignify and break down the recalcitrant components in plant cell walls. White-rot fungi have been reported to be the most effective microorganisms for decomposing lignin (*Abdel-Hamid, Solbiati & Cann, 2013*), as the basidiomycetes of white-rot fungi can efficiently degrade lignin (*Wong, 2009*) by producing various extracellular ligninolytic enzymes, such as laccase, manganese peroxidase, and lignin peroxidase (*Nagai et al., 2007*). Lignin-modifying enzymes play a critical role in degrading lignin compounds for feed suitable for ruminants (*Andrew et al., 2016*). Overwhelming evidence indicates that the combination of white rot fungi with other fungi significantly improves straw fiber degradation. In one study, wheat straw was treated with a variety of white rot fungi to improve the degradation rate of cellulose and lignin (*Arora, Sharma & Chandra, 2011*). In this study the average degradation rate of cellulose and lignin in CPS were 13.89% and 11.18%, respectively, with microbial fermentation treatment. In a separate study, oat straw degradation rates of cellulose and lignin reached 75% and 55%, respectively, when white rot fungi and filamentous fungi were mixed (*Stepanova et al., 2003*). *Schizophyllum commune* has demonstrated degradation rates of 30–40% in finger millet straw, 27–32% in paddy straw, 21% in wheat straw, and 26% in maize straw (*Kumar, Sridhar & Rao, 2022*). Moreover, carboxymethylcellulose activity was significantly enhanced when using potato peel residue as the substrate and treating the mixture with *Aspergillus Niger* and *Trichoderma* (*Taher et al., 2016*).

Using a comprehensive analysis of SEM, XRD, and FTIR, the degradation degree of lignocellulose by *Trichoderma reesei* and *Coprinus comatus* was analyzed at the microscopic level to comprehensively evaluate the effectiveness and feasibility of the fermentation method. The results show that this method can effectively degrade the lignocellulose of CPS. A previous study found, using SEM, that the process of bacteria

acting on wood can be divided into three processes: erosion, ditching, and drilling (*Rihani, Bottonb & Abbouyi, 2001*). Bacteria first grow to the middle thin layer of wood cells, causing the erosion of fiber walls, and then grow, creating cavities, or ditches, in the cell wall. Another study used XRD to confirm that the relative crystallinity of corn cobs, corn stalks, and sorghum stalks increased after ammonia was combined with ultrasonic pretreatment, likely because the pretreatment destroyed the amorphous region of lignocellulosic raw materials and exposed the crystalline region (*Xu et al., 2017b*). *Zhang et al. (2016)* used FTIR to determine the changes to chemical groups in poplar after pretreatment with white rot fungi combined with a NaOH solution, and *Xu et al. (2015)* used SEM, XRD, and FTIR to evaluate the pretreatment effect of sodium lignosulfonate on corn stalks. *Ye et al. (2016)* also used SEM, XRD, and FTIR to characterize the microstructure changes of paulownia after physical/chemical pretreatment In this study, strains producing cellulase (*Trichoderma reesei*) and laccase (*Coprinus comatus*) were used to degrade CPS, and the degradation effect of CPS was evaluated through various fermentation conditions. The results showed that these strains had a degradation effect on CPS lignocellulose.

The growth of microorganisms is influenced by several factors, including: moisture, inoculum amount, nitrogen source, carbon source, nutrient requirements, and fermentation time. Other studies have assessed the effects of pH, carbohydrates, and nitrogen source ratios in the straw fermentation process and found that adding a suitable amount of carbs and nitrogen sources during the process can significantly improve the feed fermentation quality (*Tao et al., 2017*). This study demonstrated that fungal ratio, fungal inoculum, corn flour content, and fermentation duration had important effects on the fermentation process of CPS. Increased *Coprinus comatus* led to increased lignin degradation and a synergistic degradation of cellulose. This finding explains the increase in the cellulose breakdown rate when the fungal ratio increased and the concentration of *Trichoderma reesei* decreased. When the inoculum was too large, the microorganisms grew too quickly; the insufficiently dissolved oxygen in the fermentation system reduced microorganism activity and enzyme production rate. In contrast, when the inoculum was too small, microorganism growth and the enzyme production rate were slow. Carbon and nitrogen supplies are reportedly significant factors for microbial lignin degradation and enzyme production and modifying the culture conditions can greatly increase the production of lignin-degrading enzymes (*Leontievsky, Ayasoedova & Golovleva, 1994*). Different proportions of cornmeal, used as a source of carbon and nitrogen, were tested in this study to examine their effects on CPS fermentation, and 10% cornmeal yielded optimal results. Fermentation duration also impacts CPS fermentation results. When fermentation time is too short, the bacteria cannot produce the enzymes that will degrade the straw lignocelluloses, but a fermentation time that is too long can impact the quality of the fermentation (*Zhao et al., 2020*).

A symbiosis of microorganisms can increase the palatability and digestibility of straw for livestock and poultry, reducing feed rate, costs, and environmental pollution. Research shows that fermentation with lactic acid bacteria, yeast, and *Bacillus amyloliquefaciens* increases CP content in rape straw (*Tuyen et al., 2013*). In this study, *Trichoderma reesei*

and *Coprinus comatus* were added to CPS at a 4:6 ratio, leading to different degrees of cellulose and lignin degradation the straw. More lignin was destroyed by more *Coprinus comatus*, explaining the increase in cellulose breakdown when the relative concentration of *Trichoderma reesei* declined. It is widely acknowledged that *Trichoderma reesei* and *Coprinus comatus* can synthesize laccase, manganese peroxidase, lignin peroxidase, and cellulase to degrade cellulose and lignin as quickly as possible in the solid-state fermentation process with the right inoculation dosage. However, if the inoculum is too high, the nutrients in the solid-state fermentation medium are consumed too quickly, affecting the growth of the fungal strain later in the fermentation process. At the same time, the metabolites produced increase in a short time, reducing the synthesis rate of various decomposing enzymes. The benefit of mixed fungal fermentation is that different strains exhibit different enzyme-producing properties and complement one another, resulting in the rapid breakdown of lignocelluloses in straw (*Gutierrez-Correa et al., 1999*). Laccase activity significantly improved when cultured Trametes Versicolor and oyster mushrooms were mixed with other microorganisms such as fungi, indicating that mixed cultures can achieve cellulose and lignin degradation by strengthening the extracellular lignocellulose-degrading enzymes as well as improving other active factors (*Petr, 2004*). *Trichoderma reesei* and *Coprinus comatus* were used to ferment CPS in this experiment, and the impact on cellulose degradation was the most significant after a fermentation time of 15 days. As the fermentation time increased, the remaining nutrients in the solid fermentation medium gradually decreased, the growth and metabolism of the strains slowed, and the secretion of related enzymes used for degradation decreased, resulting in a gradual decrease in the rate of cellulose and lignin degradation.

## CONCLUSIONS

The optimal fermentation conditions of *Trichodemma reesei* and *Coprinus comatus* for lignin degradation in CPS were obtained. The fungal combination fermentation could significantly increase cellulose, hemicellulose, and lignin degradation rates for converting CPS to more edible feedstuffs.

### Funding

This work was supported by the Science and Technology Project of Gansu Province (21JR1RA218); Longyuan Youth Innovation and Entrepreneurship Talent Project of Gansu Province (2021LQGR36); the Research on key technologies of efficient and healthy beef cattle breeding and comprehensive development and utilization of forage resources (LYLK201008); funds for the publication of the article were provided by the Fundamental Research Funds for the Central Universities of Northwest Minzu University (31920220038, 31920220025 and 31920230027). The funders had no role in study design, data collection and analysis, decision to publish, or preparation of the manuscript.

## Grant Disclosures

The following grant information was disclosed by the authors:
Gansu Province: 21JR1RA218.
Gansu Province: 202117.
Central Universities of Northwest Minzu University: 31920190019 and Yxm2021080.
Forage Resources: LYLK201008.

## Competing Interests

The authors declare that they have no competing interests.

## Author Contributions

- Ti Wei conceived and designed the experiments, performed the experiments, analyzed the data, prepared figures and/or tables, and approved the final draft.
- Hongfu Chen analyzed the data, prepared figures and/or tables, and approved the final draft.
- Dengyu Wu analyzed the data, prepared figures and/or tables, and approved the final draft.
- Dandan Gao performed the experiments, prepared figures and/or tables, authored or reviewed drafts of the article, and approved the final draft.
- Yong Cai performed the experiments, authored or reviewed drafts of the article, and approved the final draft.
- Xin Cao performed the experiments, authored or reviewed drafts of the article, and approved the final draft.
- Hongwei Xu performed the experiments, authored or reviewed drafts of the article, and approved the final draft.
- Jutian Yang performed the experiments, authored or reviewed drafts of the article, and approved the final draft.
- Penghui Guo conceived and designed the experiments, performed the experiments, analyzed the data, authored or reviewed drafts of the article, and approved the final draft.

## Data Availability

 The raw data is available in the Supplemental Files.

## Supplemental Information

Supplemental information for this article can be found online at http://dx.doi.org/10.7717/peerj.15757#supplemental-information.

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
