# Peer review of "Response surface methodology for the mixed fungal fermentation of Codonopsis pilosula straw using Trichoderma reesei and Coprinus comatus"

_PeerJ, doi:10.7717/peerj.15757_

## Round 0.1 · original submission · Major Revisions

Dear authors
Thank you for submitting your work to PeerJ. Keeping in mind the reviewer's comments and based on my personal review, the article is returned for MAJOR REVISIONS, the write up contains many language errors. Please revise the manuscript carefully to address all the comments in addition to review by an expert in English language. Hope to receive the revised version soon.

·

Basic reporting

This is a good study conducted by the authors; however some revisions are required prior to publication.

Experimental design

1. Scanning Electron Microscope figure should be added to study the cellulose and lignin degradation rate.
2. FTIR analysis also need to be added (Raw and degraded substrate)
3. XRD analysis also need to be done to check cellulose crystallinitysom

Validity of the findings

some graphs are missing like, desireability chart, observed and predicted values

Additional comments

1. All scientific names should be in italics
2. The article needs to be read by ta fluent English speaker
3. Writing style of the manuscript should be improved like "inoculum amount" items etc. this is not a technical writing style

Reviewer 2 ·

Basic reporting

Overall the manuscript is written in a sufficiently clear manner. The necessary background information has been included in the Introduction section but the justification for using Trichoderma reesei and Coprinus comatus specifically for this study is lacking. Please revise this.

The section on data presentation and analysis, however, does not meet the standard of PeerJ:
1. There are too many tables and figures with data duplication. Please reconsider and consolidate the tables and figures, where applicable. The data from Table 1 is being duplicated in Table 10. This is not acceptable.

2. SD/SE are missing from many tables, for instance, Table 10. Please check if the appropriate statistical analysis was performed.

3. Please consider to present some of the data in the form of bar charts or line graphs for easier visualization. For example, Tables 2-5.

Experimental design

Improvements are required for some areas:

1. Line 78: what is the method used for species authentication? Correct identification of the species used in any study is crucial.

2. Line 98: the authors stated that liquid fermentation was performed with 200 ml media in 250 flasks under shaken condition - the volume seems too high for that size? - please check

3. Lines 111-126: how did the authors selected 10 days as the baseline for fermentation?

4. Lines 141-158: please organize this section; some of the analysis, for instance, HPLC determination of organic acids, was not referenced - some basic information such as column used is needed

5. Please clarify the number of biological and/or technical replicates for this study. Similarly, consider the use of SD/SE for the mean of each treatment.

Validity of the findings

Data presented enables the authors to determine the optimal fermentation condition of Codonopsis pilosula using Trichoderma reesei and Coprinus comatus. The authors are requested to relook into data analysis and interpret their data in a more comprehensive manner - and to take into consideration previous findings on the factors affecting fermentation. The discussion, should be improved by focusing more on the data obtained rather than factual information about the organisms or fermentation process.

Additional comments

The study is interesting but the data analysis and interpretation requires improvement as per the suggestions above. I would like to see if any conclusions regarding parameters affecting the fermentation process can be drawn from this study and how the degradation rate obtained in this study fare compared to the values reported in the literature.

·

Basic reporting

The authors analyze the co-fermentation potential of Trichoderma reesei and Coprinus comatus using Codonopsis pilosula as a substrate. The authors also used RSM to validate their study. The overall scientific work is fine. However, I believe this article is lacking novelty and is an extension of already proven work. Furthermore, in the manuscript as well, the authors have not been able to highlight the significance of the study to the field. Therefore, I will suggest the authors go through the relevant literature and identify the research gaps. After identification, try to target those research gaps to highlight the novelty of their work.

The authors need to check the whole article for English as it contains many grammatical errors and sentence structurer errors.

Experimental design

Overall experimental design is satisfactory. However, there are certain information is lacking.

1. In line 84, what do you mean by Ph Natural?
2. In line 84 "The fungi were grown at 28 °C...", please clarify where you maintained this temperature (Instrument)
3. The line " In this study, the spore concentrations of the two strains were counted using a blood cell counting plate after culture to ensure spore concentration was constant (1×107 cells/mL) (Chen et al., 2018; Chen et al., 2011; Xu et al., 2017)" is not correct grammatically, Please rewrite. In addition, Hemocytiometer is a better word to use for a blood cell counting plate.
4. The amount of medium (200 ml) taken in a 250 ml flask is too much to handle while putting the flask on shaking. This will result in a false result with no productivity. This is the major technical flaw as well in this study.

Validity of the findings

1. The findings lack novelty.
2. I will suggest authors compare their work with the most recent literature (Preferably from the year 2022). This will give them a better idea of analyzing their work in terms of novelty.
3. Conclusion is good and sound.

---

## Round 0.2 · Minor Revisions

The conclusion needs to be re-written. Currently, it looks like a summary of the work rather it should state the outcome or inference of the study.
Figure 1-4, parts (a) & (b) need to be explained.

Reviewer 2 ·

Basic reporting

The authors have responded to my previous comments satisfactorily; however, clarification is needed for one response. They authors mentioned they filled 200 ml of media into 250 ml conical flasks. The volume is rather high for flasks of this capacity. Wouldn't the media spill out during the shaking process (150 pm)? Please check and confirm this.

Experimental design

No comment

Validity of the findings

No comment

---

## Round 0.3 · Minor Revisions

I have only one concern left and that is the use of 200 ml medium in 250 ml flask. I have never seen this in case of aerobic fermentation. Please review it or you may even delete the sentence mentioning the use of 200 ml medium.

---

## Round 0.4 · accepted · Accept

I would like to congratulate you on accepting your article for publication in PeerJ.
Thank you for addressing all the issues raised by the reviewers and myself and thank you for your patience throughout the review process.